# Fabrication, Structure, and Thermal Properties of Mg–Cu Alloys as High Temperature PCM for Thermal Energy Storage

**DOI:** 10.3390/ma14154246

**Published:** 2021-07-29

**Authors:** Zheng Sun, Liyi Zou, Xiaomin Cheng, Jiaoqun Zhu, Yuanyuan Li, Weibing Zhou

**Affiliations:** School of Materials Science and Engineering, Wuhan University of Technology, Wuhan 430070, China; sunzheng400@163.com (Z.S.); zouliyi@whut.edu.cn (L.Z.); zhujiaoq@whut.edu.cn (J.Z.); yyli@whut.edu.cn (Y.L.); jsyczwb@whut.edu.cn (W.Z.)

**Keywords:** magnesium-copper alloy, latent heat storage, high temperature phase change materials, thermal properties

## Abstract

This work studied the thermophysical properties of Mg-24%Cu, Mg-31%Cu, and Mg-45%Cu (wt.%) alloys to comprehensively consider the possibility of using them as thermal energy storage (TES) phase change materials (PCMs) used at high temperatures. The microstructure, phase composition, phase change temperatures, and enthalpy of these alloys were investigated by an electron probe micro analyzer (EPMA), X-ray diffraction (XRD), and differential scanning calorimetry (DSC). The XRD and EPMA results indicated that the binary eutectic phase composed of α-Mg and Mg_2_Cu exists in the microstructure of the prepared Mg–Cu series alloys. The microstructure of Mg-24%Cu and Mg-31%Cu is composed of α-Mg matrix and binary eutectic phases, and Mg-45%Cu is composed of primary Mg_2_Cu and binary eutectic phases. The number of eutectic phases is largest in Mg-31%Cu alloy. The DSC curves indicated that the onset melting temperature of Mg-24%Cu, Mg-31%Cu, and Mg-45%Cu alloys were 485, 486, and 485 °C, and the melting enthalpies were 152, 215, and 91 J/g. Thermal expansion and thermal conductivity were also determined, revealing that the Mg–Cu alloys had a low linear thermal expansion coefficient and high thermal conductivity with respect to increasing temperatures. In conclusion, the thermal properties demonstrated that the Mg–Cu alloys can be considered as a potential PCM for TES.

## 1. Introduction

Thermal energy storage (TES) is a method of storing excess heat for future use to solve the problem of an intermittent energy supply, to demand mismatch, and achieve energy balance. Due to its characteristics of environmental protection and energy saving, in accordance with the idea of sustainable development and high applicability, it has aroused great research interest in various engineering applications such as concentrated solar energy (CSP) [1], waste heat recirculation [2], building energy-saving [3], and automobile engine cooling system [4], currently some areas that are closely integrated with TES. Phase change materials (PCMs) use latent heat storage that has the characteristics of high heat storage density, small temperature fluctuation, and good reversibility of heat absorption and release processes compared with other TES methods; these materials have attracted great attention [5,6].

Thermal performance such as thermal conductivity, heat capacity, operating temperature, and thermal reliability is an important basis for selecting PCM in TES application. According to the temperature gradient, Hoshi et al. [7] subdivided the PCMs applied to the CSP system into three categories: low, medium, and high temperature PCMs. High temperature PCMs are used at temperatures above 420 °C. CSP plants require high temperature PCMs with higher operating temperatures and thermal storage density to improve system-level energy-efficiency. The candidates of high temperature PCMs generally include molten salts, eutectic mixed salt, alkalis, metals, and metallic alloys [8]. Among them, metallic alloys possess high thermal conductivity and low overcooling degree. Compared with molten salt, metal alloys usually experience less corrosion and have the potential to apply to high temperature TES.

In recent years, research on the application of Al-based binary and ternary eutectic alloys in TES has gradually aroused the interest of researchers [9,10,11]. Wang et al. [12] prepared Al-12%Si (wt.%) alloy and applied it as a TES medium in a new high temperature phase change storage space heater. Al-12%Si alloy used in various shell-tube latent thermal exchangers is determined to be an effective PCM [13,14]. Li et al. [15] synthesized Al-40%Si-15%Fe (at.%) alloy by hot pressing with a melting heat of 865 kJ/kg and a melting temperature of about 876 °C. Sun et al. [16] conducted 1000 melting and solidification cycles on Al-34%Mg-6%Zn (wt.%) alloy; results demonstrated that the alloy exhibited good thermal stability and that 304L stainless steel is a relatively suitable packaging material for Al-34%Mg-6%Zn alloy. Nevertheless, the major disadvantages of Al-based alloys is that for iron-based materials, aluminum melting is highly corrosive [17] and thus limits its durability and versatility as a candidate encapsulation material.

In terms of melting point, heat of fusion, specific heat, volume expansion coefficient, and other thermophysical properties, magnesium has properties similar to aluminum. Moreover, the Mg–Fe system maintains thermodynamic stability and immiscibility within a temperature range of 400–600 °C, indicated by the phase diagram [18]. In consideration of these factors, the use of Mg-based alloys as a new type of superalloy PCM for TES applications is promising [19]. In contrast, most studies currently are concerned about the thermal and mechanical properties of Mg–Cu binary alloys as structure materials [20,21] and there have been only a few studies on the Mg-based binary alloys such as PCM for thermal energy storage [22,23].

After comparing the different binary and ternary phase diagrams of the selected materials, the Mg–Cu system was chosen. In this work, we mainly study the thermal properties of Mg–Cu alloys as PCM at the temperature range of interest from 400–600 °C. Therefore, the Mg rich side alloys have been selected with a composition of Mg-24%Cu, Mg-31%Cu, and Mg-45%Cu (mass %). The phase composition and microstructure of test alloys were investigated. Moreover, the essential data including phase change temperature and phase-transition enthalpy, thermal expansion, and conductivity of Mg–Cu alloys were also determined for ensuing numerical simulation and experimental studies [24].

## 2. Experimental

### 2.1. Materials and Preparation

Mg–Cu alloys in weight percentage were prepared by permanent mold casting with Mg (99.98%) and Cu (99.99%) ingots (Sigma Aldrich, St. Louis, MS, USA). The chemical composition of Mg–Cu alloys determined by X-ray fluorescence (XRF, S2 PUMA, BRUKER AXS GmbH, Billerica, MA, USA) are shown in Table 1. A pit type resistance furnace (Sinopharm Chemical Reagent Co., Ltd., Shanghai, China) was used to melt 600 g materials into synthetic Mg–Cu alloys in a graphite crucible (Sinopharm Chemical Reagent Co., Ltd., Shanghai, China) with a lid (Sinopharm Chemical Reagent Co., Ltd., Shanghai, China). In order to reduce the volatilization and slag production of molten metal at high temperatures, RJ-2 flux (chemical compositions: MgCl_2_ = 40–50 wt.%, KCl = 35–45 wt.%, CaF_2_ = 5–8 wt.%, and NaCl + CaCl_2_ = 5–8 wt.%) was added to the graphite crucible in addition to Mg and Cu ingots. A high purity inert gas (99.999% argon gas, Sigma Aldrich, St. Louis, MS, USA) atmosphere was also provided to construct an oxygen-free environment that strictly prevented the samples from being oxidized during the preparation process. The melt was held at 750 °C for 30 min, then stirred at 150 rpm for 5 min, and finally poured into a cast iron mold (Sinopharm Chemical Reagent Co., Ltd., Shanghai, China) that was preheated to 200 °C with a size of inner diameter 30 mm, thickness 2 mm, and length 45 mm.

### 2.2. Analysis Methods

X-ray diffraction (XRD, D8 ADVANCE, BRUKER AXS GmbH, Billerica, MA, USA) using monochromatic Cu Kα radiation was conducted to identify the phase constitution of as-cast alloy. Electron probe micro-analysis (EPMA, JXA-8230, JEOL Ltd., Tokyo, Japan) was performed to characterize as-cast microstructures. A quantitative energy dispersive spectrometer (EDS, INCA X-ACT, JEOL Ltd., Tokyo, Japan) system attached to the EPMA instrument was used to analyze the chemical composition.

Differential scanning calorimetry (DSC, STA449F3, NETZSCH-Gerätebau GmbH, Selb, Germany) analysis was performed in a pure argon (99.999%) atmosphere with a constant heating and cooling rate of 5 °C/min from 25–510 °C. A pushrod-type dilatometer (DIL 402SE, NETZSCH-Gerätebau GmbH, Selb, Germany) was adopted for the dilatometry test. The sample was processed into a size of Ø5 mm × 23 mm to measure the linear thermal expansion coefficient and density values of alloy at the temperature range of 25–450 °C and at the heating rate of 5 K/min. The pure argon (99.999%) atmosphere was introduced into the pushrod-type dilatometer to prevent the oxidation of the sample at high temperatures. 

The laser-flash method (LFA 457, NETZSCH-Gerätebau GmbH, Selb, Germany) was used for the thermal diffusivity measurements. The samples were prepared into 8 mm × 8 mm × 2 mm blocks and the thermal diffusivity ware measured over the range of 25–450 °C under the protection of pure argon (99.999%). The test was performed with a temperature step of 50 °C and each point is tested at least 3 times. The density of the alloy at high temperatures was calculated by measuring its density at 25 °C and relative elongation at the heating temperature range through relation [25]:*ρ* = *ρ*_0_ × (1 + *ΔL/L_0_*)^−3^,(1)
where *ρ*_0_ is the density of the alloy measured by the Archimedes method at 25 °C and *ΔL/L_0_* is the relative elongation at the heating temperature. The Neumann–Kopp rules and published data were used to calculate the specific heat capacity [26]. The thermal conductivity can be calculated from the measured parameters through the following relationship:*k* = *a* × *ρ* × *c_p_*,(2)
where *a* is the thermal diffusivity, *ρ* is the density, and *c_p_* is the specific heat capacity at constant pressure.

## 3. Results

### 3.1. Structural Analysis

The X-ray diffraction patterns of as-cast Mg-24%Cu, Mg-31%Cu, and Mg-45%Cu alloys are shown in Figure 1. The sharp diffraction peaks can be identified as a mixture of hexagonal α-Mg and face-centered orthorhombic Mg_2_Cu phases, indicating that no structural change occurred in these Mg–Cu alloys. With an increased number of Cu, the diffraction intensity of the Mg_2_Cu phase are continuously strengthened and α-Mg phase is weakened.

Figure 2 reveals electron probe micro-analysis (EPMA) images of as-cast Mg–Cu alloys. Table 2 displays the composition of intermetallic phases shown in Figure 2 obtained by EDS analysis. Mg-24%Cu and Mg-31%Cu have a similar microstructure that is a typical cellular eutectic structure with dark dendrites uniformly embedded in a gray eutectic matrix. In Figure 2e,f, the light grey primary phases are embedded in the dark grey eutectic matrix.

Base on the results from the XRD and EDS analysis, it can be determined by analysis that the micron-scale primary dendrite is a α-Mg solid solution and the cellular eutectic matrix consists of a mixture of α-Mg and Mg_2_Cu in the Figure 2b,d. The light grey primary phases in Figure 2f presents Mg_2_Cu dendrites.

The solid solution (black phase) and eutectic phase (gray phase) of Figure 2 were analyzed by Image J software (Image J 1.51j8, National Institutes of Health, Bethesda, USA) and the area ratio of α-Mg + Mg_2_Cu) was calculated. Five EPMA pictures with 100× were selected for each group of samples and the average value was taken to obtain the data in the table below. For the convenience of description, Mg-24%Cu, Mg-31%Cu, and Mg-45%Cu are recorded as samples M-1, M-2, and M-3. The area of M-2 eutectic phase is the largest, M-1 is the second, and M-3 is the smallest. The area of M-1 and M-3 is 69.36%, and 42.12% is of M-2. Simultaneously, in reference to Figure 3, there is an endothermic peak at about 485 °C for each group of samples but the phase transformation enthalpy values of the samples are quite different. The maximum enthalpy value of M-2 is 215 J/g, the enthalpy values of M-1 and M-3 are 70.70%, and 40.94% corresponding to M-2. This is close to the value of area ratio, reflecting that the larger the area ratio of eutectic structure is, the greater the phase transformation enthalpy value will be. This demonstrates that the transformation enthalpy of Mg–Cu eutectic alloy has a positive correlation with the area of the eutectic phase. The area of non-eutectic and eutectic phase of alloys analyzed by Image J are listed in Table 3.

### 3.2. Phase Change Temperatures and Enthalpies

Figure 3 illustrates the DSC curves, the measured phase change temperatures, and enthalpies of Mg–Cu eutectic alloys. The DSC curves illustrates that there is only one endothermic peak during the solid-liquid transition that agrees with the phase diagram. Moreover, the fusion enthalpies of Mg-24%Cu, Mg-31%Cu, and Mg-45%Cu alloys are 152, 215, and 91 J/g, with the melting temperature of 485, 486, and 485 °C. Combining the measured melting enthalpies and the α-Mg + Mg_2_Cu eutectic content for analysis, it can be concluded that the eutectic content may be one of the main factors affecting the melting enthalpy of Mg–Cu system. This conclusion is consistent with the result of Reference [27]. Studies have shown that the content of eutectic may be one of the main factors affecting the melting enthalpy of the Mg-Cu system [27]. Combining the measured melting enthalpy and α-Mg+Mg_2_Cu eutectic content analysis, this study has reached a similar conclusion. The enthalpy of fusion of Mg-31%Cu is higher than that of Mg-24%Cu and Mg-45%Cu, and also higher than the enthalpy of fusion of the studied Mg_49_Zn_51_ (155 J/g) alloy, the eutectic phase of these three alloys proportion has a similar relationship [28].

### 3.3. Thermal Expansion

Figure 4 reveals the relative elongation and linear expansion coefficient of Mg–Cu alloys. As illustrated in the curves of Figure 4a, all the relative elongation rises with the increasing of the temperature. However, the growth rates decline with the increasing of the Cu content and downturn of the linear thermal expansion coefficient can also be observed in Figure 4b. The reason may be that Cu possesses a lower thermal expansion coefficient and is insensitive to temperature change.

The density value of Mg-24%Cu, Mg-31%Cu, and Mg-45%Cu measured experimentally at room temperature is 2063, 2207, and 2714 kg/m^3^. Figure 5 illustrates the linear fitting curve of density according to Equation (1) and the above data. The densities of the prepared alloy series are positively correlated with Cu content and negatively correlated with temperature.

### 3.4. Thermal Conductivity

Figure 6 reveals the (a) specific heat capacity and (b) thermal conductivity of Mg–Cu alloy. The specific heat capacity linearly decreases with Cu content increasing in the temperature range of 100–450 °C in Figure 6a. As Mg has a higher specific heat capacity than Cu compared to the other two alloys, Mg-24%Cu alloy displays higher specific heat capacity that passes from 0.91 kJ·kg^−1^·K^−1^ at 100 °C to 1.10 kJ·kg^−1^·K^−1^ at 450 °C. Equation (2) describes the relationship between thermal conductivity and thermal diffusivity, and between density and specific heat capacity. According to the data obtained, Equation (2) can be used to calculate thermal conductivity. As evident in Figure 6b, the thermal conductivity increases from the minimum value of 113 W·m^−1^·K^−1^ at 100 °C to the maximum value of 143 W·m^−1^·K^−1^ at 400 °C, which is much higher than previously studied Mg-36%Bi (86 W·m^−1^·K^−1^ at 100 °C, 91 W·m^−1^·K^−1^ at 400 °C) [27].

## 4. Conclusions

Mg–Cu eutectic alloy is a potential latent heat storage material that can be used for TES. This article reports the main thermal properties of Mg–Cu eutectic alloy. The binary eutectic phase composed of α-Mg and Mg_2_Cu exists in the microstructure of the prepared Mg–Cu series alloys. The microstructure of Mg-24%Cu and Mg-31%Cu is composed of a α-Mg matrix and binary eutectic phases, while Mg-45%Cu consists of the primary Mg_2_Cu and α-Mg + Mg_2_Cu eutectic. The fusion enthalpies of three alloys are 152, 215, and 91 J/g, and all the melting points are close to 485 °C. The Mg-31%Cu alloy has the highest value of melting enthalpies that can be attributed to the largest number of binary eutectic phases. Secondly, from the values of thermal expansion and thermal conductivity, the elongation, linear thermal expansion coefficient, specific heat capacity, and thermal conductivity increase with the temperature rise from 100 to 450 °C and decrease with the increasing of Cu content. The results demonstrate that the studied Mg-31%Cu alloy has the potential to be developed into PCMs for TES applications. By the same token, the compatibility of encapsulating Mg-based alloys with stainless steel will be further studied to discuss the possibility of Mg-based alloys serving as PCM.

## Figures and Tables

**Figure 1 materials-14-04246-f001:**
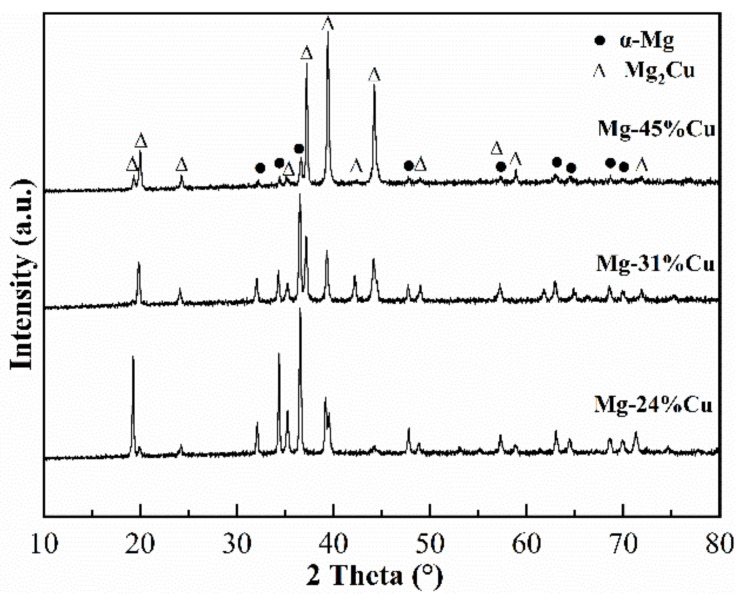
X-ray diffraction patterns of as-cast Mg–Cu alloys.

**Figure 2 materials-14-04246-f002:**
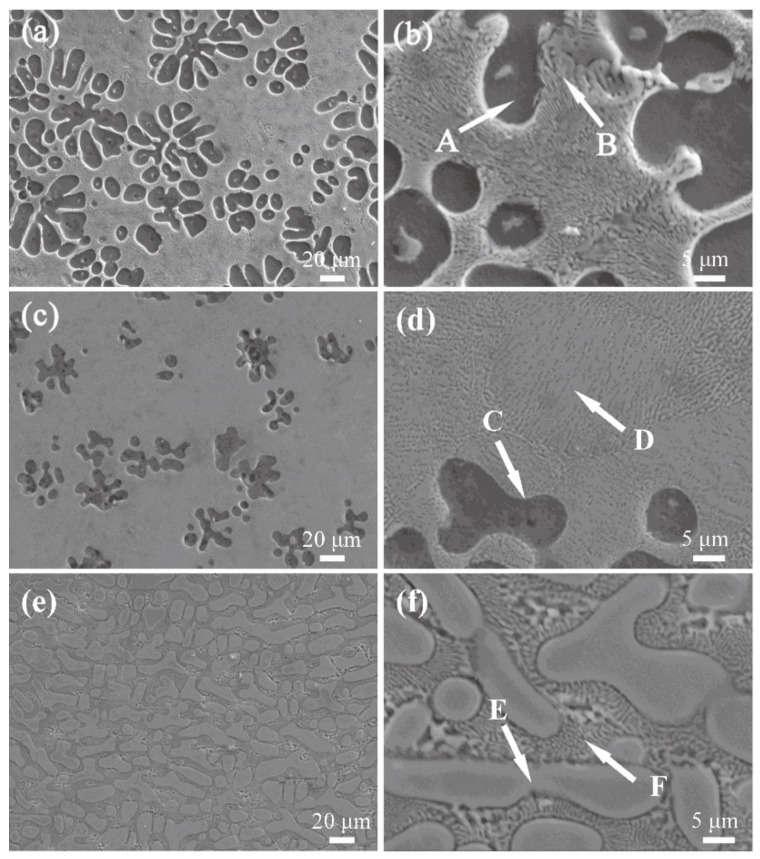
Electron Probe Micro Analysis images of as-cast Mg–Cu alloys: (**a**) Mg-24%Cu, ×400; (**b**) Mg-24%Cu, ×2000; (**c**) Mg-31%Cu, ×400; (**d**) Mg-31%Cu, ×2000; (**e**) Mg-45%Cu, ×400; and (**f**) Mg-45%Cu, ×2000.

**Figure 3 materials-14-04246-f003:**
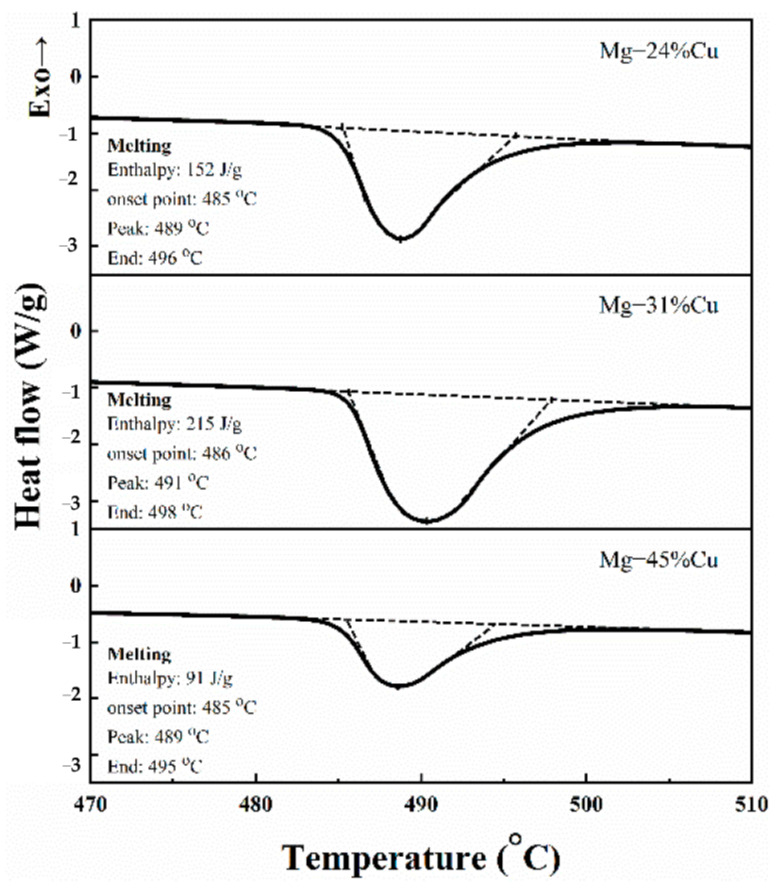
Phase change temperatures and enthalpies of Mg–Cu eutectic alloys: Mg-24%Cu, Mg-31%Cu, and Mg-45%Cu.

**Figure 4 materials-14-04246-f004:**
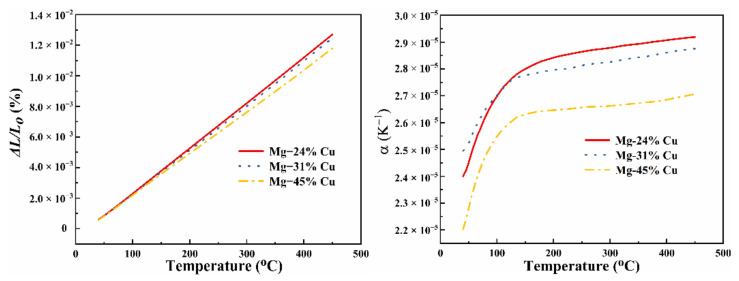
The temperature dependence of thermal expansion of Mg–Cu alloys: (**a**) relative elongation and (**b**) linear thermal expansion coefficient.

**Figure 5 materials-14-04246-f005:**
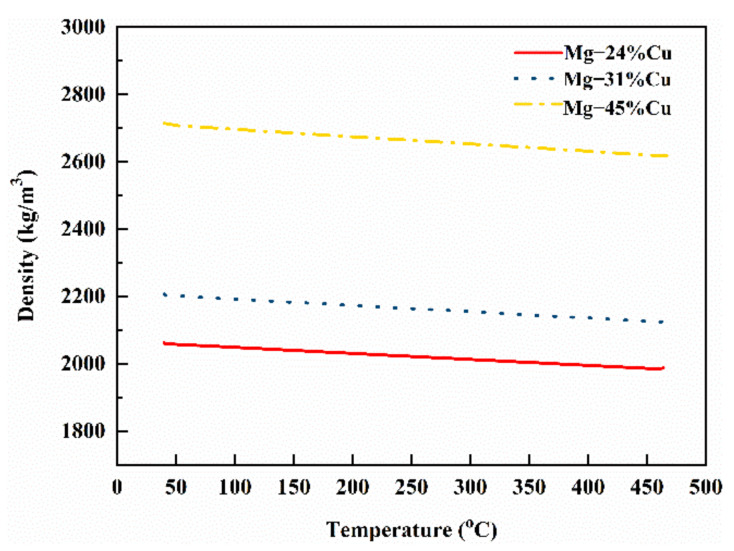
Density changes with temperature.

**Figure 6 materials-14-04246-f006:**
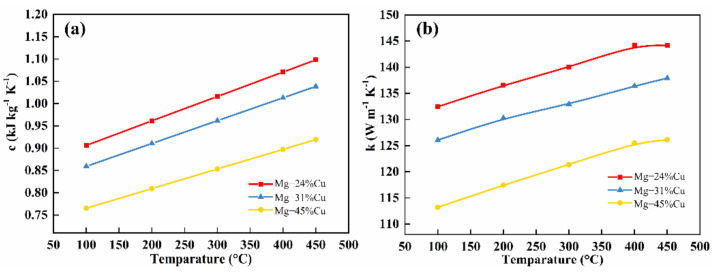
Specific heat capacity and thermal conductivity of Mg–Cu eutectic alloys: (**a**) specific heat capacity and (**b**) thermal conductivity.

**Table 1 materials-14-04246-t001:** The components of Mg–Cu alloys.

Samples	Compounds	Composition (wt.%)		
		Mg	Cu	O
M-1	Mg-24%Cu	74.91	24.37	0.72
M-2	Mg-31%Cu	67.99	31.15	0.84
M-3	Mg-45%Cu	54.50	45.10	0.40

**Table 2 materials-14-04246-t002:** Chemical composition of intermetallic phases in Figure 2 (in wt.%).

Phase	A	B	C	D	E	F
Mg	98.38	52.90	91.95	61.80	44.09	50.72
Cu	1.23	46.10	3.50	36.68	54.63	46.53
O	0.39	1.00	4.55	1.51	1.28	2.75
Closest phase	α-Mg	α-Mg + Mg_2_Cu	α-Mg	α-Mg + Mg_2_Cu	Mg_2_Cu	Mg_2_Cu + α-Mg

**Table 3 materials-14-04246-t003:** The phase transformation enthalpy and eutectic area of samples.

Samples	Compounds	Area of Non-Eutectic Phase (%)	Area of Eutectic Phase (%)	Enthalpy of Phase Transition (J·g^−1^)
M-1	Mg-24%Cu	40.627	59.373	152.0
M-2	Mg-31%Cu	14.404	85.596	215.0
M-3	Mg-45%Cu	63.943	35.040	91.0

## Data Availability

All the data is available within the manuscript.

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
