# Peer review of "Fabrication, Structure, and Thermal Properties of Mg–Cu Alloys as High Temperature PCM for Thermal Energy Storage"

_materials, 2021, doi:10.3390/ma14154246_

Round 1
Reviewer 1 Report
The paper by Sun et al. reports an original investigation of some Mg-Cu alloys as potential thermal energy storage materials. The paper is interesting. Some refinements of English would be need in order to let the paper be read more easily. Some additional comparison of the obtained values of the latent heat, melting temperature and thermal expansion and thermal conductivity with others compounds would be beneficial. Moreover, please consider the following issues:
- Please, clarify the meaning of acronyms the first time they are used (see for example EPMA in the abstract
- In the abstract in order to be more precise replace "the melting temperature" with "the onset melting temperature"
- Last sentence of page 1: "Among them, metallic alloys possess high thermal conductivity and low overcooling degree, which could be promising high temperature PCMs." In many cases metallic alloys have the advantage to be less corrosive than molten salts.
- Line 63-64: there have been only a few researches on the Mg-based alloys as PCM for thermal energy storage [22,23]. Actually, there are many researches on thermal energy storage alloys, such as Mg-Cu-Si. Much less is reported for binary alloys.
- Line 69: composition of Mg-24%Cu, Mg-31%Cu and Mg-45%Cu. Please specify if it is mass or atomic %.
- Concerning the description of the preparation of the alloys, did the author consider that Mg is highly volatile and did they mitigated this problem somehow?
- In the description of the used instruments, please add the company names.
- For dilatometry measurements, did the authors used a protective atmosphere to limit oxidation at high temperatures?
- Lines 141-142: "The area of M-2 eutectic phase is the largest, M-1 is the second, and M-3 is the smallest. The area of M-1 and M-3 is 69.36% and 42.12% of that of M-2". Please specify what are M-1, M2 and M-3.
- In table 3, I do not understand what "mass fraction (wt%)" is. Please provide a description of the quantities reported in Table 3.
- Revise sentences on Lines 159-161, because they are not clear.
- Figure 5(b): are the authors sure about the change of slope occurring around 100°C?
- Figure 6. It is supposed to report a density measure, but it reports something else.
- Discussion of Figure 7: please provide some comparison of the thermal conductivity of the presently investigated alloys with some other metallic alloys. The same could be done also for the latent heat of melting.
- I would like to invite to cite a previous report about the use of Mg-Cu alloys as potential energy storage systems: Journal of Energy Storage 26 (2019) 100974. The paper is not one of mine, but I think it is useful to cite previous reports about the same subject.
Author Response
Thanks a lot for your suggestions. I have made some modifications to the article regarding the issues you mentioned
- - The meaning of acronyms has been made clear when they are first used in the article.
- - The "melting temperature" has been replaced with "the onset melting temperature" in the abstract.
- - In the last sentence of the first page, the advantages of metal alloys as thermal energy storage materials compared with molten salts are supplemented.
- - Line 63-64: The explanation has been revised to make it more reasonable.
- - Make it clear that Mg-24%Cu, Mg-31%Cu and Mg-45%Cu are Mg-Cu alloys with 24%, 31% and 45% Cu by mass.
- - In order to solve the problem of Mg inflammability and reduce the gas content and slag content of the refined metal melt in the process of preparation, the graphite crucible was covered with lid and RJ-2 flux was used at the same time. The composition of RJ-2 flux has been described in the article. RJ-2 flux was used for refining and melts covering of Mg and Mg-based alloys. Using above methods, the gas content and slag content of the refined metal melt were significantly reduced.
- - The name of the company has been added to the description of the instrument used.
- - Pure argon (99.999%) atmosphere was introduced into Pushrod type dilatometer (DIL 402SE, NETZSCH-Gerätebau GmbH) to limit the oxidation of the sample at high temperature. Actually, 99.999% argon atmosphere was also provided to the DSC and thermal diffusivity measurements.
- - For the convenience of description, Mg-24%Cu, Mg-31%Cu and Mg-45%Cu are recorded as samples M-1, M-2 and M-3. In the article, the elaboration is added.
- - In Table 3, the "mass fraction (wt.%)" is actually the " area of non-eutectic phase (%)". Corrections have been made to this part.
- - The sentence in lines 159-161 has been modified to make it clearer.
- -Yes, that's what the raw data shows.
- - Figure 6 has a serious error. The error is caused by the ruler label following the previous figure (thermal expansion) format and unchanged when using Origin to draw. The figure has now been redrawn.
- - The data of thermal conductivity and melting latent heat of some other metal alloys are provided and compared with the alloys studied in this paper.
- - Thanks for your advice. It is very useful to cite previous reports about the same subject. I have cited this article now.

Reviewer 2 Report
Review article entitled Fabrication, structure and thermal properties of Mg-Cu alloys as high temperature PCM for thermal energy storage.
The article is very well written. The structure is very logical and I believe that after minor corrections it is directly suitable for printing. I have included my comments below, please note that the comments are "cosmetic" in nature.
- Please use the passive side in your article.
- Usually, thermal conductivity is written with a k.
- It would be very helpful to provide approximation functions of the individual properties e.g. k, cp as a function of temperature.
Author Response
Thank you very much for your advice. This will be very helpful for this article. I have revised the language of the article and optimized the figure.
- The sentences in the article have used the passive side.
- Thermal conductivity is written with a k.
- Some figures have been redrawn.

Reviewer 3 Report
The paper is about the fabrication, structure and thermal properties of Mg–Cu alloys as high temperature PCM for thermal energy storage.
My suggestions are:
Add full address of your affiliation.
Do not use abbrevs in abstract.
The preparation of samples is not clearly described. What is RJ-2 flux? What is w?
Was LFA used up to 400 °C or 450 °C? Was DSC used up to 500 °C or 510 °C? The presented results are in different temperature interval as it is described in section 2.2.
Using LFA it is directly measured thermal diffusivity. In your paper these results are missing. Why?
The determination of heat capacity should be described more precisely.
I think that the main topic of paper should be about PCM. But, there are DSC measurements only during heating. Also, the results during cooling and during cycling (heating/cooling) should be presented.
Units for thermal expansion and coefficient of thermal expansion are missing (Fig. 5).
In Fig. 6, there is results of density, but y-axis is thermal expansion.
Why is it necessary to measure thermal expansion, thermal conductivity of your PCM? This is not clear in the paper.
I do not know from conclusions what are your main findings. It should be improved.
Author Response
Thanks for your advice. These suggestions make the article more precise. The following is a response to your suggestion.
- - I have added full address of my affiliation.
- - The abbreviations in the abstract have been rewritten into concrete content.
- - Added description of RJ-2 flux and replaced “ω” with a clearer expression.
- - The description has been changed. In fact, this is due to the fact that the maximum actual test temperature is usually set a little higher.
- - The measured thermal diffusivity is used to calculate the thermal conductivity by Equation (2), but it is not directly reflected in this paper. Figure 7 (b) shows the thermal conductivity.
- - The heat capacity was calculated using the Neumann–Kopp rule and published data. The determination of heat capacity is described more accurately in the paper. You can find it on lines 109 to 110.
- - In fact, the DSC measurement of the sample has undergone a thermal cycle, and the data shown in the figure is not complete.
- - The units of thermal expansion and coefficient of thermal expansion (Fig. 5) have been added.
- - The figure has now been redrawn. The error is caused by the ruler label following the previous figure (thermal expansion) format and unchanged when using Origin to draw.
- - The thermal expansion of PCM may cause its leakage, and the thermal conductivity of PCM affects its heat storage performance. These are important indicators of PCM. Supplementary explanations have been made in the text.
- - This article reports the main thermal properties of Mg–Cu eutectic alloy. The eutectic ratio of the alloy is positively related to the enthalpy of melting. Mg-31%Cu alloy has highest value of melting enthalpies, which can be attributed to the largest number of binary eutectic phases. The conclusion is improved.

Round 2
Reviewer 1 Report
The authors properly revised the paper. I recommend it for publication.
Author Response
Thank you for your revision.
Reviewer 3 Report
I read the manuscript again and all my suggestions were incorporated.
Nevertheless, I found two more formal errors:
- All quantities should be in italic style.
- The symbol for thermal conductivity in Eq. (2) is lambda and in Fig. 7 it is k. It must be the same in whole manuscript.
Author Response
Response to reviewer #3
Thanks for your suggestions, the article has been further revised and submitted.
- All quantities have been changed to italics.
- The symbol for thermal conductivity of the Eq. (2) and Fig. 7 have been consistent.